# RoSe: Rotation-Invariant Sequence-Aware Consensus for Robust Correspondence Pruning

Yizhang Liu
School of Software Engineering, Tongji University
College of Computer and Data Science, Fuzhou University
China
lyz8023lyp@gmail.com

Weiwei Zhou
School of Software Engineering, Tongji University
Shanghai, China
zhouweiwei@tongji.edu.cn

Yanping Li
Department of Computer Science and Technology, Tongji University
Shanghai, China
liyp8023yz@tongji.edu.cn

Shengjie Zhao
School of Software Engineering, Tongji University
Engineering Research Center of Key Software
Technologies for Smart City Perception and Planning,
Ministry of Education
Shanghai, China
shengjiezhao@tongji.edu.cn

## ABSTRACT

Correspondence pruning has recently drawn considerable attention as a crucial step in image matching. Existing methods typically achieve this by constructing neighborhoods for each feature point and imposing neighborhood consistency. However, the nearest-neighbor matching strategy often results in numerous many-to-one correspondences, thereby reducing the reliability of neighborhood information. Furthermore, the smoothness constraint fails in cases of large-scale rotations, leading to misjudgments. To address the above issues, this paper proposes a novel robust correspondence pruning method termed RoSe, which is based on rotation-invariant sequence-aware consensus. We formulate the correspondence pruning problem as a mathematical optimization problem and derive a closed-form solution. Specifically, we devise a rectified local neighborhood construction strategy that effectively enlarges the distribution between inliers and outliers. Meanwhile, to accommodate large-scale rotation, we propose a relative sequence-aware consistency as an alternative to existing smoothness constraints, which can better characterize the topological structure of inliers. Experimental results on image matching and registration tasks demonstrate the effectiveness of our method. Robustness analysis involving diverse feature descriptors and varying rotation degrees further showcases the efficacy of our method.

## CCS CONCEPTS

• **Computing methodologies → Matching**; **Visual content-based indexing and retrieval**; **Epipolar geometry**.

## KEYWORDS

Correspondence pruning, Rotation-invariant, Sequence-aware

**ACM Reference Format:**
Yizhang Liu, Weiwei Zhou, Yanping Li, and Shengjie Zhao. 2024. RoSe: Rotation-Invariant Sequence-Aware Consensus for Robust Correspondence Pruning. In *Proceedings of the 32nd ACM International Conference on Multimedia (MM '24), October 28-November 1, 2024, Melbourne, VIC, Australia.* ACM, New York, NY, USA, 10 pages. https://doi.org/10.1145/3664647.3681124

## 1 INTRODUCTION

Establishing accurate correspondence is crucial for many computer vision tasks, including image registration [7, 28, 40, 49], image fusion [16, 41], image retrieval [17, 18], 3D reconstruction [1, 46], and simultaneous localization and mapping (SLAM) [37, 39, 43]. However, the reliability of feature descriptors of matching images can be compromised by various factors, such as illumination changes, scale transformations, and the presence of noise. For instance, feature points extracted by the SIFT [29] algorithm often exhibit similar descriptors in repetitive structures. This similarity can lead to a substantial number of incorrect correspondences (outliers) when initial correspondences are determined based on nearest-neighbor criteria. Consequently, the implementation of effective and robust correspondence pruning algorithms is of great significance and has garnered considerable attention.

Due to the diversity of transformations and challenges between images, existing parametric methods such as RANSAC [11], as well as learning-based approaches, fail to achieve satisfactory results across a wide range of general scenarios. Specifically, RANSAC-related methods fall short in dealing with non-rigid transformations, and they often struggle to produce good results within a reasonable time when faced with a high proportion of outliers among initial correspondences. Learning-based methods [12, 13, 19, 30, 52], on the other hand, cast the correspondence pruning problem into a binary classification task, while incorporating the geometric constraints (e.g., essential matrix loss) to boost model performance. However, this strategy limits the network's ability to generalize to other types of transformations.

Recently, motion consistency-based methods have shown promising performance in general correspondence pruning scenarios. These methods are typically based on an intuitive observation that feature points belonging to the same structure exhibit similar motion behaviors [20, 21, 26, 27], thus they are general enough and can be applied to various scenarios. For such methods, the reliability of local neighborhood information of feature points and how motion consistency is formulated have a significant impact on the model performance. However, the widely used nearest-neighbor matching criterion often results in many-to-one correspondences, where at most only one correspondence is inlier. Therefore, when feature points involved in many-to-one correspondences contribute to neighborhood construction, there is a significant negative impact on the local neighborhood information of the feature points. In addition, in cases of large-scale rotations, inliers within a small region may exhibit substantial differences in length and angle, leading to the breakdown of smoothness constraints.

To address the aforementioned challenges, this paper introduces a novel mathematical model solely based on rotation-invariant sequence-aware consensus, which is simple yet effective and general. Specifically, to mitigate the impact of many-to-one correspondence on feature point neighborhood construction, we propose a rectified local neighborhood construction strategy. Initially, we identify many-to-one correspondences, and subsequently exclude feature points involved in these correspondences during neighborhood construction for each feature point. Statistics on publicly available datasets confirm that this strategy effectively expands the distribution between inliers and outliers, thereby facilitating correspondence pruning. Additionally, to address large-scale rotations, we introduce the concept of relative sequence-aware consistency as a substitute for conventional smoothness constraints. This is grounded in the assumption that the relative ordering of inliers is preserved post-transformation. We ascertain this by calculating the longest common subsequence length from the ordered sequences of the $k$ nearest neighbors of matched feature points. Notably, the rectified local neighborhood construction also provides a solid foundation for relative sequence-aware consistency by enlarging the length of the ordered sequences of the $k$ nearest neighbors. In short, our method effectively handles reliable neighborhood construction and large-scale rotations and generalizes well to other challenges and transformations. The contributions of this paper are summarized as follows:

1. To the best of our knowledge, this paper is the first to investigate the effects of many-to-one correspondences on the neighborhood construction of feature points. We propose a rectified local neighborhood construction strategy, effectively enlarging the distribution between inliers and outliers.

2. We develop a novel rotation-invariant relative sequence-aware consistency that is both versatile and more effectively captures the topological structure of inliers during large-scale rotations.

3. Experiments on image matching and registration across a range of publicly available datasets demonstrate optimal or comparable performance. Further, our method's robustness is highlighted through validation studies that involve distinct feature descriptors and different rotation degrees.

## 2 RELATED WORK

Existing correspondence pruning methods are mainly categorized into traditional methods and learning-based methods.

### 2.1 Traditional Correspondence Pruning

RANSAC [11] is one of the representative resampling-based methods, which estimates model parameters by randomly sampling a subset of the original data and calculating the number of inliers that fit the model. This process is repeated multiple times, ultimately selecting the model with the highest number of inliers as the best fit. Despite enhancements in RANSAC through optimizing sampling strategies and termination criteria, etc. [2–4, 8], these methods still encounter challenges with sensitivity to high outlier ratios and inefficiencies in handling non-parametric transformations. Motion consistency-based methods [31] generally operate on the assumption that inliers within a local region exhibit similar motion behaviors, albeit using different formulations to construct distinct models based on motion consistency. LPM [34] introduces two types of consistencies: neighborhood element consistency and neighborhood topology consistency, which are utilized to ensure that the unknown inlier correspondences possess similar local neighborhood structures. RFMSCAN [15] proposes to cluster correspondences with similar neighborhood elements and topologies by using DBSCAN. LAF [14] treats outliers as noise, and solves the correspondence pruning in remote-sensing images using a linear adaptive filtering method. LOGO [42] proposes a locality-guided strategy to guide global information optimization. A reliable subset with high ratio inliers is determined by local topology consistency, which is then progressively expanded to yield the final inlier set. The performance of these methods heavily relies on the accuracy with which the local neighborhood structures of feature points are constructed. In addition, under large-scale rotations, most existing consistency constraints become ineffective, leading to a significant degradation in model performance. Technically, our method belongs to the same category as LPM, but it enhances the reliability of the neighborhood construction for feature points. Meanwhile, it introduces a rotation-invariant relative sequence-aware consensus, making it broadly applicable and general.

### 2.2 Learning-based Correspondence Pruning

Given the widespread adoption of deep learning across diverse domains, correspondence pruning has evolved into a binary classification problem employing deep models. LFGC [44] pioneers a deep learning model to accomplish correspondence pruning. It draws inspiration from the highly successful PointNet framework in point cloud processing, utilizing a shared-weight Multi-Layer Perceptron (MLP) architecture and permutation-invariant operations to extract the global context of correspondences. Building on this, OANet [47] introduces the DiffPool and DiffUnpool layers to learn the local context of correspondences, and construct an Order-Aware Filtering block to capture the global context of correspondences. MS2DGNet [9] proposes to construct multiple sparse semantics dynamic graphs to capture local topology among correspondences. PGFNet [22] devises an iterative filtering structure, wherein the outcomes of one iteration guide the network learning in the subsequent iteration, further improving the reliability of contextual information. CLNet [51]

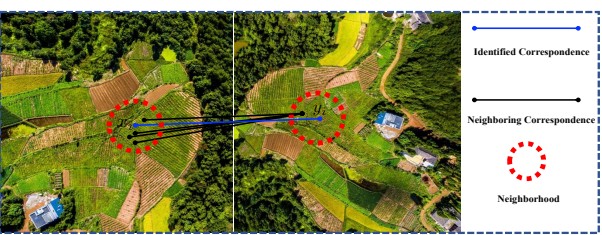

**Figure 1: The impact of many-to-one correspondences on the neighborhood construction of feature points. At most only one of the four neighboring correspondences is an inlier, resulting in unreliable statistics of neighborhood information for feature points.**

introduces a pruning framework that progressively learns local and global consistency scores for correspondence pruning. Model estimation is then conducted by selecting correspondences with high global consistency scores. Building on CLNet [51], NCMNet [23] proposes to construct $k$-nearest neighbors for each correspondence within graph space, combining existing coordinate-space $k$-nearest neighbors and feature-space $k$-nearest neighbors to create a more comprehensive local context for correspondences. Furthermore, ConvMatch [48] presents a novel architecture that employs convolutions to learn and aggregate local features for correspondences. While these methods incorporate the essential matrix loss in the loss function to improve model performance, this often results in decreased adaptability to other transformations. Additionally, learning-based methods heavily rely on training data and typically exhibit weaker generalization capabilities across different datasets.

## 3 PROPOSED METHOD

### 3.1 Rectified Local Neighborhood Construction

Given two matching images $I_1$ and $I_2$, feature point detection and description are conducted, resulting in $N$ and $M$ feature points for each image, respectively. Using the nearest-neighbor matching strategy, the $N$ feature points in $I_1$ are paired with their closest counterparts among the $M$ feature points in $I_2$, forming $N$ correspondence, denoted as $S = \{(x_i, y_i)\}_{i=1}^{N}$. Previous methods determine the correctness of correspondences by constructing local neighborhoods for feature points and analyzing the consistency among their neighboring correspondences. Intuitively, when two feature points are close in the image space, they probably share the same structure. Consequently, they are expected to remain close after transformation [6, 45, 50]. In other words, if $(x_i, y_i)$ is an inlier, its neighborhood usually contains inliers that mutually support each other. Conversely, if it is an outlier, its neighborhood is likely sparse with correspondences, attributed to the random distribution of outliers. The neighborhood element consistency is widely utilized in correspondence pruning methods [15, 24, 25, 34] and has been proven to be highly effective in certain scenarios.

To guarantee the efficacy of neighborhood element consistency, it is essential to ensure that the local neighborhoods of feature points are predominantly composed of inliers [33]. Existing methods construct the neighborhoods for feature points $x_i$ and $y_i$ by

identifying their $k$ nearest neighbors among all feature points in images $I_1$ and $I_2$, respectively. However, this straightforward approach to constructing neighborhoods can include a large number of outliers. Given that feature points in images $I_1$ and $I_2$ are detected and described independently, this often leads to $N \neq M$ in most cases. When $N$ significantly exceeds $M$, the initial correspondence set $S$ is prone to include numerous many-to-one correspondences, namely, different $x_i$ correspond to the same $y_i$. Suppose $x_1, x_2, \cdots, x_n$ correspond to the same $y_i$, but among these $n$ correspondences, at most one is inlier. Under these circumstances, the neighborhood information of feature points becomes extremely unreliable. We illustrate an example in Figure 1, where for correspondence $(x_i, y_i)$, we calculate the $k$-nearest neighbors for both $x_i$ and $y_i$ (with $k = 4$). As depicted in Figure 1, when the $k$-nearest neighbors of $x_i$ align with those of $y_i$, there are 4 neighboring correspondences within the neighborhood of $(x_i, y_i)$. Ideally, if these neighboring correspondences are all inliers, then $(x_i, y_i)$ is likely to be an inlier as well. However, since all $k$-nearest neighbors of $x_i$ correspond to the same neighbor of $y_i$, in practice, most of the neighboring correspondences are outliers, which makes the neighborhood construction of feature points unreliable.

Existing motion consistency-based methods have ignored the impact of many-to-one correspondences on the neighborhood construction of feature points, thereby limiting their performance. We conduct a study about the proportion of many-to-one correspondences in initial correspondences for the urban change detection dataset. As shown in Figure 2, we observe that in the majority of image pairs, the proportion of many-to-one correspondences exceeds half, highlighting the prevalence and significance of this issue.

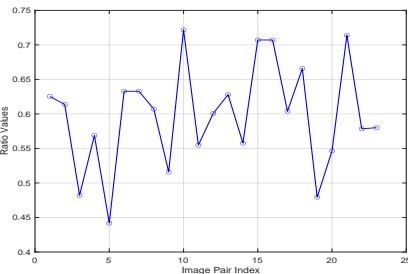

**Figure 2: Proportion of many-to-one correspondences in initial correspondences for the urban change detection datasets**

To address the aforementioned issue, we propose a simple yet highly effective strategy named rectified local neighborhood construction. Intuitively, since many-to-one correspondences inherently contain outliers, excluding feature points involved in these correspondences from neighborhood construction can significantly increase the number of potential inliers within these neighborhoods. To investigate the impact of this strategy on the local neighborhood information of correspondences, we perform statistical experiments on the urban change detection datasets. We analyze the disparity in the number of neighboring correspondences of inliers and outliers at $k = 20$, contrasting the original neighborhood construction strategy and our proposed rectified local neighborhood construction strategy. As shown in Figure 3, our rectified strategy effectively

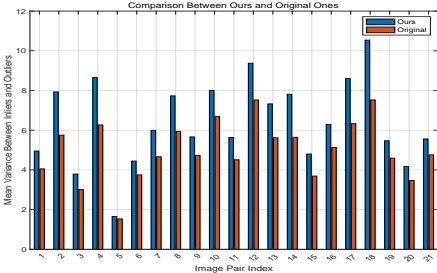

**Figure 3: On the urban change detection dataset, we compare the difference in the number of neighboring correspondences of inliers and outliers at $k = 20$ when using the original neighborhood construction strategy and our rectified local neighborhood construction strategy.**

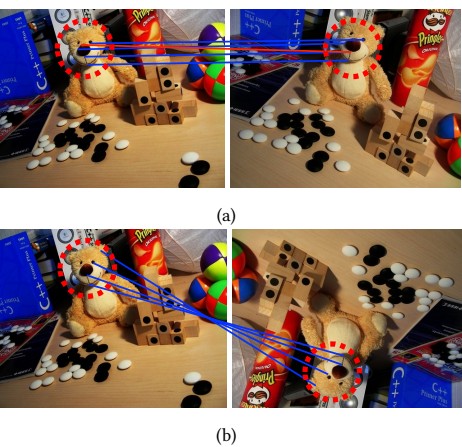

**Figure 4: (a) fails to differentiate between inliers and outliers based solely on neighborhood element consistency. (b) illustrates the invalidation of motion smoothness.**

enlarges the distribution between inliers and outliers, facilitating the correspondence pruning.

### 3.2 Relative Sequence Consistency

While neighborhood element consistency helps eliminate clear outliers, it operates on a coarse-grained level that ignores the topological structure of the neighborhood elements, thereby limiting its effectiveness in further identifying inliers. To more clearly illustrate this problem, we provide an example in Figure 4a. The local neighborhoods of $x_i$ and $y_i$ are completely identical, perfectly satisfying the neighborhood element consistency. Consequently, $(x_i, y_i)$ is identified as an inlier, whereas $(x_i, y_i)$ is actually an outlier. Existing methods primarily tackle this issue by measuring the differences in length and angle between $(x_i, y_i)$ and its neighboring correspondences, which is a kind of motion smoothness constraint. Within a local scope, inliers exhibit smoothness (e.g., only referring to translation transformation). When matching images are subject to wide baselines or large-scale rotations, the smoothness constraint

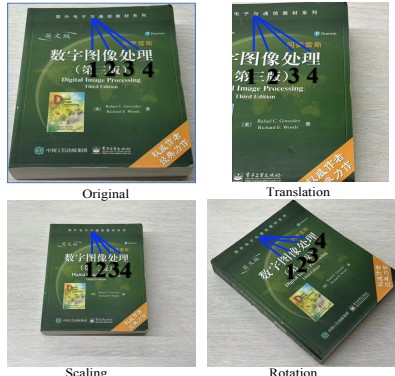

**Figure 5: The relative sequence of inliers does not change freely with translation, scaling, and rotation.**

becomes ineffective. As shown in Figure 4b, we can observe that the vector fields formed by inliers are not smooth, with noticeable variations in both their lengths and angles. Consequently, distinguishing inliers from outliers under these conditions proves to be a challenging task.

To address the above issue, we exploit the inherent property that the local structure of objects will not change freely after image transformations to propose a novel consistency named rotation-invariant relative sequence-aware consistency. Specifically, we denote the $k$-nearest neighbors ordered list for feature point $x_i$ as $\sigma(x_i) = [x_i^1, x_i^2, ..., x_i^k]$, and for feature point $y_i$ as $\sigma(y_i) = [y_i^1, y_i^2, ..., y_i^k]$. The neighboring correspondences of the correspondence $(x_i, y_i)$ are defined as $\{(x_c, y_c)|x_c, y_c \in \sigma(x_i) \cap \sigma(y_i)\}$. Whenever the number of neighboring matches is nonzero, we consider two overlapping ordered lists $\phi(x_i)$ and $\phi(y_i)$.

$$\phi(x_i) = [x_c|x_c \in \sigma(x_i) \cap \sigma(y_i)], \quad (1)$$

$$\phi(y_i) = [y_c|y_c \in \sigma(x_i) \cap \sigma(y_i)], \quad (2)$$

Obviously, $\phi(x_i)$ and $\phi(y_i)$ contain exactly the same elements but not necessarily in the same sequence. Under ideal rigid transformations, if $(x_i, y_i)$ is an inlier, its local neighborhood is entirely made up of inliers, satisfying $\sigma(x_i) = \phi(x_i)$, $\sigma(y_i) = \phi(y_i)$, and $\phi(x_i) = \phi(y_i)$. This is due to the inherent physical constraints of the object, which ensure that inliers maintain a fixed relative positional relationship. As shown in Figure 5, we observe that regardless of translation, scaling, or rotations involved in the matching images, the relative sequence of the inliers remains unchanged. Therefore, we achieve relative sequence-aware consistency by measuring the length of the longest common subsequence between $\phi(x_i)$ and $\phi(y_i)$. The longest common subsequence, which is not necessarily contiguous, serves as a robust measure of similarity between two ordered lists by quantifying the extent of shared elements, which focuses on the relative sequence rather than the absolute sequence. We determine the length of the longest common subsequence using the Dynamic Programming (DP) algorithm.

Given that the lengths of $\phi(x_i)[j]$ and $\phi(y_i)[k]$ are $n$, we first initialize an $n \times n$ zero matrix $L$, which serves as a repository for solutions to all subproblems. As we proceed to iteratively examine the elements within $\phi(x_i)$ and $\phi(y_i)$, the value of $L[j, k]$ is updated

contingent on the equality of $\phi(x_i)[j]$ and $\phi(y_i)[k]$. If the two values are equal, it means these two elements can be part of the longest common subsequence, so the value of $L[j, k]$ is set to the value of its top-left neighbor plus 1. Conversely, in instances of inequality, $L[j, k]$ adopts the greater value between its left neighbor $L[j-1, k]$ and its upper neighbor $L[j, k-1]$. Finally, the element in the bottom-right corner of the matrix, $L(n, n)$, is the length of the longest common subsequence. The specific steps of the algorithm are summarized in Algorithm 1.

---

**Algorithm 1** Pseudocode for computing the length of the longest common subsequence.

---

**Input:** Two ordered lists $\phi(x_i)$, $\phi(y_i)$
**Output:** Length of the longest common subsequence $L[n, n]$
 1: Create a matrix $L_{n \times n}$ and initialize to 0
 2: **for** $j$ from 1 to $n$ **do**
 3:     **for** $k$ from 1 to $n$ **do**
 4:         **if** $\phi(x_i)[j] == \phi(y_i)[k]$ **then**
 5:             $L[j, k] = L[j-1, k-1] + 1$
 6:         **else**
 7:             $L[j, k] = max(L[j-1, k], L[j, k-1])$
 8:         **end if**
 9:     **end for**
10: **end for**
11: **return** $L[n, n]$

---

## 3.3 Problem Formulation and Solution

Let $n_i$ denote the number of neighboring correspondences for $(x_i, y_i)$, and let $l_i$ signify the length of the longest common subsequence determined from these correspondences. Denote the unknown inlier set as $\mathcal{I}$, and the correspondence pruning problem can then be formulated as the following optimization problem:

$$\mathcal{I}^* = \arg \min_{\mathcal{I}} C(\mathcal{I}; S, \lambda), \tag{3}$$

where $\mathcal{I}^*$ denotes the optimal inlier set, and $C$ is the cost function:

$$C(\mathcal{I}; S, \lambda) = \sum_{i \in \mathcal{I}} \left( \frac{(k - n_i)}{k} + \beta \frac{(n_i - l_i)}{n_i} \right) + \lambda(N - |\mathcal{I}|), \tag{4}$$

where $k$ represents the number of nearest neighbors associated with feature points $x_i$ and $y_i$. $\beta$ is the weighted coefficient balancing the neighborhood element consistency and the relative sequence-aware consistency.

The first term of the cost function aims to penalize the correspondences that violate neighborhood element consistency and relative sequence-aware consistency. The second term serves as a regularization measure to prevent model overfitting, with $\lambda$ as the balancing coefficient balancing these two terms. We introduce an $N \times 1$ binary vector $\mathbf{p}$ to indicate the correctness of correspondences, where $p_i = 1$ indicates that $(x_i, y_i)$ is an inlier, and $p_i = 0$ points to an outlier. Therefore, the cost function mentioned above

can be rewritten as:

$$C(\mathbf{p}; S, \lambda) = \sum_{i=1}^{N} p_i \left( \frac{(k - n_i)}{k} + \beta \frac{(n_i - l_i)}{n_i} \right) \tag{5}$$
$$+ \lambda \left( N - \sum_{i=1}^{N} p_i \right).$$

By merging the terms associated with $p_i$, we can obtain:

$$C(\mathbf{p}; S, \lambda) = \sum_{i=1}^{N} p_i(c_i - \lambda) + \lambda N, \tag{6}$$

where

$$c_i = \sum_{i=1}^{N} \left( \frac{(k - n_i)}{k} + \beta \frac{(n_i - l_i)}{n_i} \right). \tag{7}$$

Observing Eg. 6, it is evident that $c_i > \lambda$ will lead to an increase in the cost function, whereas $c_i < \lambda$ will result in a decrease. Thus, determining the correctness of correspondences can be achieved through the following simple way:

$$p_i = \begin{cases} 1, c_i \leq \lambda \\ 0, c_i > \lambda \end{cases}, i = 1, \ldots, N. \tag{8}$$

The optimal inlier set $\mathcal{I}^*$ is represented as follows:

$$\mathcal{I}^* = \{(x_i, y_i) | p_i = 1, i = 1, ..., N\}. \tag{9}$$

## 3.4 Implementation Details

To ensure the reliability of feature point neighborhood construction, we adopt an iterative strategy similar to LPM [34]. In the first iteration, neighborhood constructions for each correspondence are based on the initial correspondence set $S$, and $c_i$ for each correspondence is computed according to Eq. 7. The potential inliers can be obtained according to Eq. 8. The optimal inlier set in this iteration is denoted as $S_1$. In the second iteration, neighborhoods for each correspondence are constructed using $S_1$ instead of $S$, as $S_1$ contains a higher inlier ratio compared to $S$, enhancing the reliability of the feature point neighborhood construction. Similarly, $c_i$ is calculated using Eq. 7, further using Eq. 8 to obtain the optimal inlier set $\mathcal{I}^*$. In both iterations, the number of nearest neighbors $k$ is set to 20. The weighted coefficients $\beta$ for balancing neighborhood element consistency and relative sequence-aware consistency are both set to 1. $\lambda$ is set to 0.15 in the first iteration and 0.35 in the second iteration.

## 4 EXPERIMENTS

### 4.1 Datasets and Evaluation Metrics

*4.1.1 Datasets.* We evaluate our method on the following three datasets and compare it with state-of-the-art methods: 1) Remote Sensing (RS) dataset [32], which encompasses four types of remote sensing images, i.e., aerial images, synthetic aperture radar images, panchromatic aerial photographs, and infrared color aerial photographs. The dataset consists of 161 image pairs, featuring challenges such as projection/affine distortions, small overlap, repetitive structure, and high outlier ratio challenges. 2) Urban Change Detection (UCD) dataset [10], which consists of 21 multispectral image

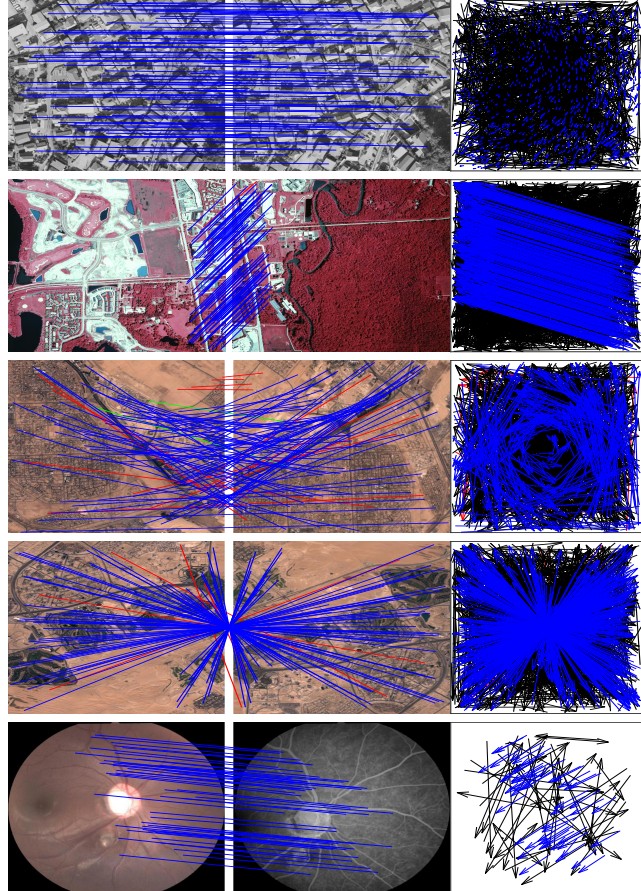

**Figure 6: Qualitative image matching results of our RoSe on five representative image pairs: PAN90, CIAP129, UCD160, UCD169, and Retinal14. Each group includes the image matching results along with their corresponding motion vector fields. The head and tail of each arrow in the motion vector field indicate the positions of the feature points in the two images (blue = true positive, black = true negative, green = false negative, red = false positive). For clarity, we randomly select up to 100 correspondences to display in each image pair, true negatives are not shown.**

pairs, featuring challenges such as low resolution, repetitive structures, and large-scale rotations. 3) Medical Retina (MR) dataset [16], which comprises 70 multimodal retinal image pairs obtained using various angiographic techniques, containing slight non-rigid deformations.

*4.1.2 Evaluation Metrics.* For image matching, the performance is evaluated by Precision, Recall, and F-score. The F-score serves as a comprehensive measure of matching efficacy. A larger F-score indicates better matching performance. For image registration, the performance is evaluated by Root Mean Square Error (RMSE), Maximum Error (MAE), and Median Error (MEE).

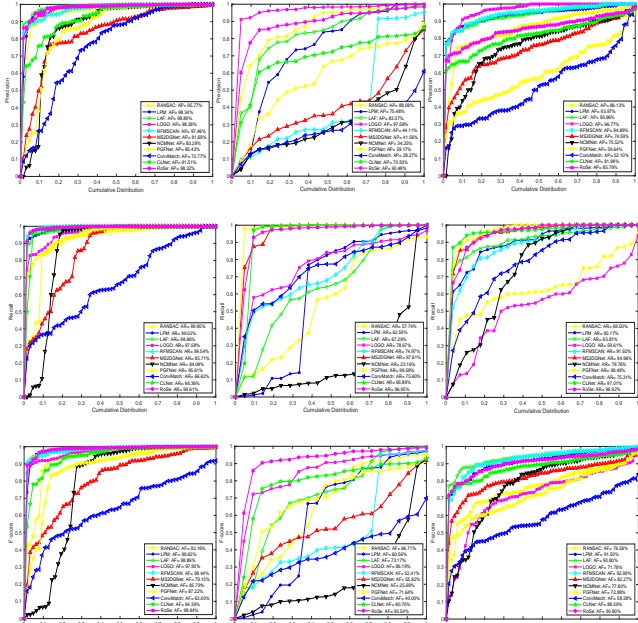

**Figure 7: Performance statistics of image matching on three datasets. From left to right, RS, UCD, and MR datasets. The coordinate $(x, y)$ on the curve indicates that there are $(100^*x)\%$ image pairs whose Precision/Recall/F-score is no more than $y$.**

## 4.2 Image Matching

We select five traditional methods: RANSAC [11], LPM [34], LAF [14], LOGO [42], RFMSCAN [15] and five learning-based methods: MS2DGNet [9], NCMNet [23], PGFNet [22], ConvMatch [48], CLNet [51] for comparative analysis.

*4.2.1 Qualitative Results.* We first show the image matching results of our RoSe on five representative image pairs: PAN90, CIAP129, UCD160, UCD169, and Retinal14. PAN90 is characterized by significant repeated structures, causing strong ambiguity in feature descriptors and leading to a low initial inlier ratio of 12.8%. CIAP129 features small overlapping areas, also resulting in a low initial inlier ratio of 10.1%. UCD160 and UCD169 involve rotations at varying scales. Retinal14, with a limited number of initial correspondences (156), presents a challenge for identifying consistency among inliers. As shown in Figure 6, our RoSe demonstrates good performance in all five image pairs, validating the effectiveness and generalization ability of the algorithm.

*4.2.2 Quantitative Results.* We further conduct a quantitative comparison with state-of-the-art methods on the aforementioned datasets to demonstrate the efficacy. As shown in Figure 7, it is evident that our RoSe achieves the best recall and F-score for the RS dataset while maintaining comparable precision. For the MR dataset, although our method exhibits lower precision than traditional methods, it achieves higher recall. For the comprehensive performance metric, the F-score, our results are relatively favorable. The lower precision

of our method can be attributed to i) the uniform threshold setting used throughout the experiments (based on the RS dataset), without adjustments for each specific dataset, leading to a precision-recall imbalance. ii) The initial correspondence number for each image pair in the MR dataset is relatively low, which is unfavorable for our method to exploit the inherent consistency of inliers. Notably, our RoSe demonstrates significant advantages over other competitors for the UCD dataset, leading the second-best method by 7.35% in the F-score. Except for LOGO [42], most methods struggle with large-scale rotations. For RANSAC [11], rotation is a rigid transformation that can be characterized by a parametric model. However, due to the low inlier ratio of the correspondences in this dataset, it is hard to find the optimal transformation. Among the learning-based methods, CLNet [51] performs best, benefiting from its simpler architecture that potentially reduces overfitting risks but suffers from training data bias, affecting generalization. Overall, our method exhibits outstanding performance across various transformations and challenges, particularly with large-scale rotations.

Table 1: Quantitative results of image registration. RMSE: root mean square error; MAE: maximum error; MEE: median error; RT: run time. The best results are boldfaced.

| Method | RMSE | MAE | MEE | RT(s) |
|---|---|---|---|---|
| RANSAC [11] | 25.627 | 222.912 | 2.6506 | 2.4421 |
| LPM [34] | 2.1767 | 49.6659 | 0.0004 | **0.0172** |
| LAF [14] | 2.5412 | 51.3587 | 0.0004 | 0.0632 |
| LOGO [42] | 6.7671 | 134.262 | 0.0004 | 0.1998 |
| RFMSCAN [15] | 2.8510 | 51.7178 | 0.0004 | 0.1293 |
| MS2DGNet [9] | 9.0689 | 153.880 | 0.0004 | 0.7904 |
| NCNMet [23] | 83.136 | 347.564 | 86.520 | 0.7392 |
| PGFNet [22] | 9.4205 | 161.137 | 0.0004 | 0.7182 |
| ConvMatch [48] | 70.931 | 434.225 | 22.379 | 1.1486 |
| CLNet [51] | 10.902 | 178.060 | 0.0004 | 0.5021 |
| RoSe | **1.5866** | **34.8641** | **0.0004** | 0.2494 |

to [14], obtaining the identified inliers, TPS (Thin Plate Spline) [35] is selected to estimate the transformation function. Then, for each pixel in the sensed image, its corresponding coordinates in the reference image can be calculated through the transformation function, and the intensity at that coordinate can be calculated using a bicubic interpolation algorithm.

*4.3.1 Qualitative Results.* We first show the qualitative image registration visualization comparison of all methods on the SAR58 image pair, as shown in Figure 8. SAR58 features severe noise and low texture. For most methods, they exhibit noticeable distortions in the registration results except LAF [14] and RoSe. Most traditional methods successfully recover the valid transformation function. By contrast, many learning-based methods yield strange registration results, primarily due to their limited generalization capability, which is induced by the imposition of the essential matrix loss. Since the rectified local neighborhood construction significantly enhances the reliability of neighborhoods for correspondence and aids in the rotation-invariant sequence-aware consistency, our Rose can identify most of inliers, thereby recovering accurate transformation function and producing high-quality registration results.

*4.3.2 Quantitative Results.* We further provide quantitative comparison results of all methods. From Table 1, it can be seen that traditional methods such as LPM [34], LAF [14], and RFMSCAN [15] perform well in terms of RMSE, MAE, and MEE metrics. LOGO [42], while demonstrating low RMSE, exhibits high MAE values, indicating its registration performance is not stable. For learning-based methods, MS2DGNet [9], PGFNet [22], and CLNet [51] also show low RMSE but high MAE values, indicating their instability. The overall registration performance of the remaining methods is inadequate. In contrast, our method significantly outperforms all state-of-the-art methods in terms of RMSE, MAE, and MEE metrics. This demonstrates that our RoSe can retain more inliers in a global scope, which is beneficial for accurately estimating transformations between images. Regarding runtime, LPM [34] is the most efficient, whereas RANSAC [11] is the least efficient. Our method builds on LPM [34] and improves by integrating new designs to ensure more inliers are retained, which is relatively time-consuming.

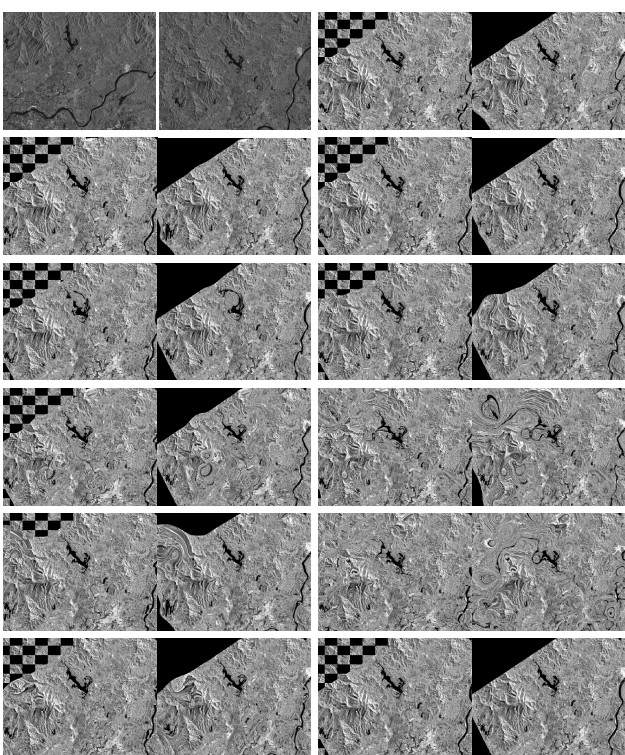

Figure 8: Qualitative image registration results. (From top to bottom and left to right) Original image pair, RANSAC [11], LPM [34], LAF [14], LOGO [42], RFMSCAN [15], MS2DGNet [9], NCMNet [23], PGFNet [22], ConvMatch [48], CLNet [51] and RoSe on the SAR58.

## 4.3 Image Registration

We choose the same state-of-the-art methods as the image matching task for comparison. The registration experiments are conducted on the RS dataset, which poses a variety of challenges. Similar

**Table 2: Image matching results for the initial correspondences obtained by different descriptors. The best results are boldfaced.**

| Method | VGG-SIFT | | | VGG-SURF | | | VGG-ORB | | |
|---|---|---|---|---|---|---|---|---|---|
| | Precision | Recall | F-score | Precision | Recall | F-score | Precision | Recall | F-score |
| RANSAC [11] | 0.9511 | 0.9806 | 0.9643 | 0.8468 | **0.9974** | 0.9136 | 0.8671 | **0.9970** | 0.8979 |
| LPM [34] | 0.9710 | 0.8550 | 0.8687 | 0.9434 | 0.7933 | 0.8053 | **0.9724** | 0.9156 | 0.9420 |
| LAF [14] | 0.9712 | 0.9683 | 0.9692 | 0.9048 | 0.9525 | 0.9244 | 0.9479 | 0.9697 | 0.9580 |
| LOGO [42] | 0.9612 | 0.9560 | 0.9586 | 0.9102 | 0.9106 | 0.9068 | 0.9020 | 0.9044 | 0.8979 |
| RFMSCAN [15] | 0.9424 | 0.9498 | 0.9437 | 0.8202 | 0.9727 | 0.8863 | 0.9007 | 0.9792 | 0.9344 |
| MS2DGNet [9] | 0.9007 | 0.8777 | 0.8735 | 0.7389 | 0.9755 | 0.8251 | 0.7468 | 0.9874 | 0.8367 |
| NCMNet [23] | 0.9504 | 0.9616 | 0.9539 | 0.7636 | 0.8504 | 0.7917 | 0.7860 | 0.8589 | 0.8125 |
| PGFNet [22] | 0.9472 | 0.9059 | 0.9119 | 0.7874 | 0.9874 | 0.8603 | 0.8249 | 0.9835 | 0.8859 |
| ConvMatch [48] | 0.9237 | 0.6126 | 0.7197 | 0.6736 | 0.7438 | 0.6649 | 0.6680 | 0.7332 | 0.6532 |
| CLNet [51] | 0.9595 | **0.9889** | 0.9694 | 0.8189 | 0.9519 | 0.8728 | 0.8504 | 0.9987 | 0.9155 |
| RoSe | **0.9789** | 0.9729 | **0.9758** | **0.9445** | 0.9748 | **0.9623** | 0.9523 | 0.9739 | **0.9614** |

**Table 3: Images matching results for the initial correspondences built at different rotation scales. The best results are boldfaced.**

| Method | 30° | | | 60° | | | 90° | | |
|---|---|---|---|---|---|---|---|---|---|
| | Precision | Recall | F-score | Precision | Recall | F-score | Precision | Recall | F-score |
| RANSAC [11] | 0.8373 | 0.7023 | 0.7789 | 0.8117 | 0.7366 | 0.7612 | 0.8423 | 0.7402 | 0.7928 |
| LPM [34] | 0.8282 | 0.9525 | 0.8816 | 0.8104 | 0.9568 | 0.8724 | 0.8071 | 0.9560 | 0.8702 |
| LAF [14] | 0.8912 | 0.9179 | 0.9026 | 0.8447 | 0.8739 | 0.8551 | 0.8211 | 0.7812 | 0.7958 |
| LOGO [42] | **0.9698** | 0.5910 | 0.7137 | **0.9753** | 0.5948 | 0.7168 | **0.974** | 0.5875 | 0.7105 |
| RFMSCAN [15] | 0.9010 | 0.7931 | 0.8330 | 0.8566 | 0.6406 | 0.7176 | 0.7797 | 0.5519 | 0.6228 |
| MS2DGNet [9] | 0.6771 | 0.9470 | 0.7790 | 0.6094 | 0.9340 | 0.7181 | 0.5559 | 0.9122 | 0.6681 |
| NCNMet [23] | 0.6922 | 0.6313 | 0.6383 | 0.5784 | 0.4213 | 0.4645 | 0.5182 | 0.3051 | 0.3614 |
| PGFNet [22] | 0.6111 | **0.9936** | 0.7385 | 0.5617 | **0.9873** | 0.7006 | 0.5378 | **0.9939** | 0.6809 |
| ConvMatch [48] | 0.4915 | 0.7080 | 0.5541 | 0.4544 | 0.7006 | 0.5312 | 0.4475 | 0.6430 | 0.5000 |
| CLNet [51] | 0.8255 | 0.9537 | 0.8819 | 0.8269 | 0.9470 | 0.8782 | 0.7997 | 0.8985 | 0.8429 |
| RoSe | 0.8579 | 0.9692 | **0.9080** | 0.8579 | 0.9692 | 0.9080 | 0.8579 | 0.9692 | **0.9080** |

## 4.4 Robustness Analysis

*4.4.1 Different Descriptors.* To validate the robustness of RoSe for initial correspondences generated by different descriptors, we conduct experiments on the VGG dataset [36]. We utilize SIFT [29], SURF [5], and ORB [38] descriptors to create initial correspondences, and report the results in Table 2. It is apparent that except for LAF [14], LOGO [42], CLNet [51], and our RoSe, the other methods exhibit significant differences or inadequate performance for different descriptors. For instance, LPM [34] achieves an F-score of 0.9420 on VGG-ORB, but only reaches 0.8687 on VGG-SIFT. NCM-Net [23] achieves an F-score of 0.9539 on VGG-SIFT, but drops to 0.7917 on VGG-SURF. Our RoSe demonstrates exceptional robustness, achieving optimal F-values on VGG-SIFT, VGG-SURF, and VGG-ORB, consistently surpassing 0.96, thus showcasing the strong robustness of our method for different descriptors.

*4.4.2 Different Rotation Scales.* To validate the robustness of RoSe for different rotation scales, for image pairs on the MR dataset, we keep one fixed and rotate the other by 30°, 60°, and 90° to create initial correspondences. From Table 1 and Table 3, it can be observed that as the rotation scale increases, there is a noticeable decline in performance for LAF [14], RFMSCAN [15], MS2DGNet [9], NCMNet [23], PGFNet [22] and ConvMatch [48] indicating that these method lack rotational invariance. RANSAC [11], being

a parametric model, it is robust to rotation, but its performance is limited by the initial inlier ratio. LOGO [42] performs poorly on the MR dataset across various rotation scales. LPM [34] and CLNet [51] exhibit robustness to rotation and achieve good matching results. Our RoSe shows exceptional robustness to rotations at different scales, maintaining consistent performance for all rotation scales, and demonstrating the rotation-invariant property.

## 5 CONCLUSION

This paper introduces RoSe, a novel and robust correspondence pruning method. RoSe comprises two innovative components: a rectified local neighborhood construction and a rotation-invariant relative sequence-aware consistency. The first component significantly mitigates the impact of many-to-one correspondences in the neighborhood construction of feature points, thereby enhancing the distinction between inliers and outliers. The second component offers a general and fine-grained consistency that possesses both rotation-invariant and sequence-aware attributes. Both qualitative and quantitative experimental results from image matching and registration tasks underscore the superior performance of our method in facing diverse challenges. The robustness analysis further demonstrates that our method effectively adapts to various descriptors and showcases outstanding rotational invariance.

## 6 ACKNOWLEDGMENTS

This work was supported in part by the National Key R&D Program of China 2023YFC3806000 and 2023YFC3806002, in part by the National Natural Science Foundation of China under Grant 61936014, in part by Shanghai Municipal Science and Technology Major Project No. 2021SHZDZX0100, and in part by Shanghai Science and Technology Innovation Action Plan Project 22511105300.

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
