# OpenReview forum: "RoSe: Rotation-Invariant Sequence-Aware Consensus for Robust Correspondence Pruning"
_acmmm.org/ACMMM/2024/Conference — MM2024 Oral_

### Official Review · Reviewer_TSM5 · 2024-05-13

**Rating:** 3
**Confidence:** 4

**Summary:**

This paper presents RoSe, a robust correspondence pruning method for image matching. Correspondence pruning is a crucial step in image matching, aiming to remove incorrect correspondences (outliers) and retain accurate ones (inliers). The proposed RoSe method addresses two main issues in existing methods: the unreliability of neighborhood information due to many-to-one correspondences and the failure of smoothness constraints in large-scale rotations.
To mitigate the impact of many-to-one correspondences, RoSe introduces a rectified local neighborhood construction strategy, which effectively enlarges the distribution between inliers and outliers, facilitating correspondence pruning. Furthermore, to handle large-scale rotations, RoSe proposes a rotation-invariant relative sequence-aware consistency that better captures the topological structure of inliers.
The paper formulates the correspondence pruning problem as a mathematical optimization problem and derives a closed-form solution. The authors demonstrate the effectiveness of RoSe through experimental results on image matching and registration tasks using various publicly available datasets. The method's robustness is also analyzed by validating its performance under different feature descriptors and rotation scales. Overall, the paper contributes to image matching and registration by proposing a simple yet effective and general method for robust correspondence pruning.

**Strengths:**

This work is based on a mathematical optimization problem and derives a closed-form solution. It is simple and elegant, which is very important for real-time applications; I would like to see its implementation, e.g., SfM and SLAM. I hope it could provide python packages instead of MATLAB.

**Limitations:**

This paper targets a standard computer vision problem; the main limitation is that this topic is narrowed to ACM MM. There are two folds to explain it:
1. This topic is focused on Correspondence Pruning, which is usually a component included in the matcher. For example, https://github.com/cvg/LightGlue, "LightGlue: Local Feature Matching at Light Speed" ICCV23. This means that Correspondence Pruning and matching are highly related, simplifying the matcher to the nearest-neighbor matching (Ln221) may detach from practical applications;
2. This topic is fine for ICCV and CVPR (I guess it has just been rejected from CVPR24), but it is relatively less important for MM. From my experience, even if the reviewers and AC agree to accept this paper, PC still may remove it from the track cause so many CV papers are submitted to the conference.

In this case, you have to prove your work is excellent to others. The original experiment seems to work with nearest-neighbor matching; this is a standard way to prove it works. But for real applications, you should show some applications that work with modern Matcher and contribute to, e.g., SLAM and SfM.
I suggest you give a Python package instead of Matlab; this may help others implement your code in many other applications. I am concerned if it is working in standard SfM, SLAM pipeline, like "https://github.com/colmap/colmap","https://github.com/colmap/pycolmap","https://github.com/introlab/rtabmap".
I am looking into the code to see if it can be implemented and improved upon works.
If you can give an implementation, it will be nice. A pull request or GitHub project may be, but I understand this may need time. It may not be completed in the rebuttal period.

**Suitability:**

2

---

### Official Review · Reviewer_rSUc · 2024-05-18

**Rating:** 5
**Confidence:** 3

**Summary:**

This paper proposes a new method termed RoSe for correspondence pruning. In contrast to most of the previous consistency-based methods that neglect the negative impact of many-to-one correspondences on the neighborhood construction of feature points, the proposed method devises a rectified local neighborhood construction strategy to ensure reliable neighborhood construction for feature points, effectively enlarging the distribution between inliers and outliers. Meanwhile, to ensure rotation-invariant property, a relative sequence-aware consistency is proposed, serving as an alternative to existing smoothness constraints, which can better characterize the topological structure of inliers. The method is evaluated on several datasets and outperform other state-of-the-art methods.

**Strengths:**

1. The paper is well-written and easy to read. It provides detailed insights into its individual contributions and overall approach to the problem.
2. The relation and difference to existing methods are also well discussed.
3. The motivation and idea of this paper are clearly stated, which is intuitive. Specifically, this paper presents a groundbreaking discovery regarding the impact of many-to-one correspondences on the neighborhood construction of feature points. Through a simple yet effective rectified strategy, the proposed method significantly expands the distribution between inliers and outliers. The rectified local neighborhood construction is a notable advanced technique that could have strong potential for all motion consistency-based correspondence pruning methods. Furthermore, the relative sequence-aware consistency outperforms traditional smoothness consistency that can struggle with large-scale rotation, which is promising and robust to rotation, translation, and scaling.
4. The experimental design is rigorous and thoughtfully constructed, providing a solid foundation for the results presented. The qualitative and quantitative image matching and registration results show the efficacy of the method. The rotation-invariant property of the method has been proven under different rotation scales. Results using distinct feature descriptors also show the robustness of the method.

**Limitations:**

Several comments are given below to further Improve the paper.
1. In the abstract, two novel components are proposed, namely, a rectified local neighborhood construction strategy and a relative sequence-aware consistency. To maintain terminological consistency, it is recommended that the title of Section 3.2 be revised to 'Relative Sequence-Aware Consistency'.
2. The rectified local neighborhood construction strategy primarily addresses the impact of many-to-one correspondences on the neighborhood construction of feature points. However, can such many-to-one correspondences be mitigated through the application of mutual nearest neighbor matching?
3. Based on the analysis presented in Figure 5 and the methodology outlined in Algorithm 1, the proposed relative sequence-aware consistency demonstrates robustness against transformations such as translation, scaling, and rotation. Consequently, the term 'Rotation-Invariant' in the paper's title could be replaced with 'Versatile' to more accurately reflect the model's adaptability.
4. In Table 3, the highest F-score results at a rotation scale of 60 are not boldfaced.
5. In Figure 8, the registration distortions associated with various methods can be distinctly marked by employing red boxes for clear labelling.

Some recent related literature needs to be discussed in the related work section. Such as:
PT-Net: Pyramid transformer network for feature matching learning.
Multi-stage network with geometric semantic attention for two-view correspondence learning.
Robust heterogeneous model fitting for multi-source image correspondences.

**Suitability:**

3

---

### Official Review · Reviewer_kC8A · 2024-05-23

**Rating:** 4
**Confidence:** 3

**Summary:**

This paper introduces a novel method called RoSe for robust correspondence pruning, which addresses two main issues in existing correspondence pruning methods:
- The nearest-neighbour matching strategy often results in many-to-one correspondences, which can reduce the reliability of neighbourhood information.
- The smoothness constraint, commonly used in existing methods, fails in cases of large-scale rotations, leading to misjudgments.
To tackle these problems, RoSe introduces:
- A rectified local neighbourhood construction strategy that effectively enlarges the distribution between inliers and outliers.
- A rotation-invariant relative sequence-aware consistency, which better characterizes the topological structure of inliers, especially under large-scale rotations.

**Strengths:**

1. The authors investigate the effect of many-to-one correspondence on the neighbourhood construction of feature points and propose a rectified neighbourhood construction strategy.
2. This paper proposes a novel rotationally invariant relative sequence-aware consistency to compute the matching consistency using the longest common subsequence (LCS) algorithm, and experimental results show that it is indeed effective.

**Limitations:**

1. More detailed comparisons in more extensive datasets would have been more convincing. It would be beneficial to have quantitative comparisons that include other relevant metrics.
2. There is no detail in Section 3.1 on how the rectified local neighbourhood construction strategy works, and more detail is needed.
3. The paper does not mention any ablation studies that would help to understand the impact of the rectified local neighbourhood construction and the rotation-invariant relative sequence-aware consistency.
4. The paper focuses on the robustness of RoSe to large-scale rotations. Still, it does not explain how it performs with other transformations such as scaling, shearing, or non-rigid deformations.

**Suitability:**

2

---

### Meta-Review · Area_Chair_56nV · 2024-07-03

**Recommendation:** Accept (Oral)
**Confidence:** 5

**Metareview:**

This paper proposes a new method for image correspondence pruning. On the one hand, this work deals with a core problem at the heart of many computer vision tasks; on the other hand, the problem of matching points between two images is not at the heart of the ACM MM community's priorities. Nevertheless, according to the 3 confident reviewers, the contribution is technically good, well-explained and fully experimented (including the rebuttal experiments) on relevant benchmarks facing a large panel of state-of-the-art recent techniques. The ablation study part is consistent and correctly discussed, and the rebuttal of the authors globally respond to the reviewers expectations. This work deserves to be published here, depending on the ratio of "monomodal" articles to be accepted at ACM MM'24.